?Amphictis (Carnivora, Ailuridae) from the Belgrade Formation of North Carolina, USA

Baskin Jon jon.baskin@retiree.tamuk.edu 1 2
Dickinson Edwin 3
DuBois John 4
Galiano Henry 5
Hartstone-Rose Adam 3
1 Department of Biological and Health Sciences, Texas A&M University—Kingsville , Kingsville , TX , United States of America
2 UT Jackson School of Geoscience, University of Texas at Austin , Austin , TX , United States of America
3 Department of Biological Sciences, North Carolina State University , Raleigh , NC , United States of America
4 Benson , NC , United States of America
5 Maxilla & Mandible, Ltd. , New York , NY , United States of America
Thewissen J.g.m.
Electronic publication date: 2020 Jul 8
Publication date: 2020
Volume: 8
Electronic Location ID: e9284
Received 2020 Mar 18; Accepted 2020 May 12
Copyright: ©2020 Baskin et al.
Copyright year: 2020
Copyright holder: Baskin et al.
License: This is an open access article distributed under the terms of the Creative Commons Attribution License, which permits unrestricted use, distribution, reproduction and adaptation in any medium and for any purpose provided that it is properly attributed. For attribution, the original author(s), title, publication source (PeerJ) and either DOI or URL of the article must be cited.
License URL: https://creativecommons.org/licenses/by/4.0/

Keywords: ?Amphictis, Carnivora, Ailuridae, Belgrade formation, North Carolina

Funding: The authors received no funding for this work.

==============================
Miocene terrestrial mammals are poorly known from the Atlantic Coastal Plain. Fossils of the Order Carnivora from this time and region are especially rare. We describe a carnivoran mandible with a p4 from the late Oligocene or early early Miocene Belgrade Formation in Jones County, North Carolina. Comparisons are made with carnivoran jaws with similar premolar and molar lengths from the late Oligocene and Miocene of North America and Eurasia. These indicate that the North Carolina jaw is assignable to the Ailuridae, a family whose only living member is the red panda. The jaw is tentatively referred to Amphictis, a genus known elsewhere from the late Oligocene and early Miocene of Europe and the early Miocene (Hemingfordian) of North America.

The North Carolina mandible compares best with the late Oligocene (MP 28) Amphictis ambiguus from Pech du Fraysse, France, the oldest known member of the Family Ailuridae, and with the early Miocene (MN 1–MN 2a) A. schlosseri from southwestern Germany. This identification is compatible with a late late Arikareean (Ar4, early Miocene, MN 2-3 equivalent) age assignment for the other terrestrial mammals of the Belgrade Formation.

Introduction

Tertiary terrestrial mammals are known from a few localities on Atlantic Coastal Plain (Tedford et al., 2004). Only four of these contain terrestrial carnivorans. The Farmingdale Local Fauna of New Jersey, initially assigned to the early Hemingfordian (Tedford & Hunter, 1984), is late late Arikareean (Ar4, MN 2–3 equivalent) in age (Emry & Eshelman, 1998). An indeterminate canid is recorded from there (Gallagher et al., 1995). The early Hemingfordian (He1, MN 4 equivalent) Pollack Farm Local Fauna of Delaware is from the Calvert Formation. It contains at least 26 species of land mammals (Emry & Eshelman, 1998). Carnivorans present are two procyonids, two borophagine canids, an ursid, and two amphicyonids. The Barstovian Chesapeake Bay Fauna of Maryland and Virginia is from the Calvert and overlying Choptank Formations (Wright & Eshelman, 1987). Amphicyon and the borophagine canid ?Cynarctus marylandica are recorded in the fauna (Tedford et al., 2004). The Lee Creek Mine Local Fauna is from the Lower to Upper Pliocene Yorktown Formation in North Carolina (Snyder, Mauger & Akers, 1983). Eshelman & Whitmore Jr (2008) documented 16 taxa from there, including two borophagine canids, an ursid, and a felid.

The Belgrade Quarry in Onslow and Jones Counties, North Carolina exposes strata of Oligocene through Pleistocene age (Locality 20 of Ward, Lawrence & Blackwelder, 1978: fig. 2; Locality 1 of Zullo, 1984). At the quarry, marine invertebrates and vertebrates have been recovered from the River Bend Formation and the overlying Belgrade Formation (e.g., Rossbach & Carter, 1991; Boessenecker, Ahmed & Geisler, 2017). The River Bend Formation is a shell hash limestone. The Belgrade Formation, which is divided into the lower Pollocksville and upper Haywood Landing Members, consists of marine shell beds, sands, and clay (Ward, Lawrence & Blackwelder, 1978).

The carnivoran jaw (NCSM 33670) described below is likely from the Belgrade Formation. At the quarry, the middle to late Oligocene River Bend Formation is mined for aggregate (McLeod & Barnes, 2008). Belgrade Formation sediments are bulldozed off of it and placed into small piles that are exposed to the elements. A mammal jaw with a p4 (NCSM 33670) was recovered while surface collecting for shark teeth and other fossils after a heavy rainstorm. Ray dental plates and shark teeth from these spoil piles include Aetobatis sp., Dasyatis sp., Myliobatis sp., Rhinoptera sp., Galeocerdo sp., Hemipristis serra, Carcharhinus sp., Carcharodon angustidens, Carcharias sp., Nebrius sp., Isurus sp., and Physogaleus sp.

The age of the Belgrade Formation is disputed. It has been assigned a late Oligocene age (Rossbach & Carter, 1991; Harris & Laws, 1997; Ward, 2002), a late Oligocene/early Miocene age (Ward, 1998), and an early Miocene age (Baum, Harris & Zullo, 1978; Zullo, Willoughby & Nystrom Jr, 1982). The Belgrade Formation in Onslow County, including the designated type section (Baum, Harris & Zullo, 1978), has been assigned to the early Miocene (Ward, Lawrence & Blackwelder, 1978; Zullo, 1984).

Terrestrial vertebrates recovered from mine spoil piles are likely earliest Miocene (late late Arikareean, Ar4, 18-23 Ma, MN 2-3 equivalent) in age (MacFadden et al., 2017). The key specimen for this determination is the entelodont Daeodon. The holotype of Daeodon leidyanus is from the late late Arikareean Farmingdale Local Fauna. However, the range of Daeodon in North America is early Arikareean to early Hemingfordian (Lucas, Emry & Foss, 1998). Pliocene mammal fossils are also present at Belgrade Quarry (B MacFadden, pers. comm., 2019). Therefore, it is necessary to compare NCSM 33670 with late Oligocene to Pleistocene taxa.

Materials & Methods

Microtomography scans of the mandible were collected on a Zeiss Xradia 510 Versa nanoCT system, housed at the Analytical Instrumentation Facility at North Carolina State University. Measurements of fossil mammals at the American Museum of Natural History were made with digital calipers to the nearest 0.1 mm. Subdivision of the Arikareean North American Land Mammal Age (Ar1-4) follows Albright et al. (2008).

Results

Systematic paleontology

Order CARNIVORA Bowdich, 1821	
Family AILURIDAE Gray, 1843	

Remarks

Baskin (1998b) had previously grouped the musteloids with an elongate m2 into a single family, the Procyonidae, with two subfamilies (Procyoninae and Ailurinae). His Ailurinae contained the tribes Simocyonini and Ailurini. In the present paper, we follow the classification of Morlo & Peigné (2010) for the Ailuridae. Ailuridae is the sister taxon to the Procyonidae and Mustelidae. Ailurus fulgens, the red panda, is the only extant ailurid. The late Oligocene to early Miocene Amphictis is the sole genus of the paraphyletic Amphictinae (Morlo & Peigné, 2010) and is the stem sister taxon to the other two ailurid subfamilies, the Ailurinae and Simocyoninae.

Genus AMPHICTIS Pomel, 1853

Genotypic species: Amphictis antiqua (De Blainville, 1842)

Remark

Morlo & Peigné (2010) diagnosed the genus and characterized the eight species of Amphictis they considered valid: A. antiqua (De Blainville, 1842); A. ambigua (Gervais, 1872); A. milloquensis (Helbing, 1928); A. borbonica (Viret, 1929); A. schlosseri (Heizmann & Morlo, 1994); A. wintershofensis Roth in (Heizmann & Morlo, 1994); A. prolongata (Morlo, 1996); and A. cuspida (Nagel, 2003). Baskin (2017) added a ninth species, A. timicua.

?AMPHICTIS sp.

Referred specimen. NCSM 33670, right mandible with p4, and alveoli for p1-p3, m1-m2; from Belgrade Quarry in Jones County, North Carolina, USA.

Description: The mandible extends from posterior to the canine to the posterior alveolus of m2 (Fig. 1). It is somewhat shallower beneath the p3 (Table 1). Posterior from beneath the p3, the ventral border is gently convex posterior to beneath the p3. The mental foramen is below the anterior root of the p3, at approximately half the depth of the ramus. The coronoid process is missing. Length-wise cracks on the surface, porous bone, and polish on parts of the surface are indications of weathering and perhaps transport before burial.

Figure 1 Lateral (A & B), occlusal (C & D) and medial (E & F) views of volumetric reconstruction of NCSM 33670.

Mandibular bone (in gray) has been rendered semitransparent in B, D, and F to display root morphology. Preserved dental elements (root and crown fragments) are tan. Alveolar casts are rendered in red (canine), blue (p1), pink (p3), green (m1) and purple (m2). A–D, anterior is to the right; E & F, anterior is to the left. Scale bar = 1 cm.

Table 1 Measurements in mm on NCSM 33670.

p2 L	5.7	
p3 L	6.1	
p4l	8.1	
p4 W	4.0	
m1 L	10.9	
m2 L	6.9	
p2-m1 L	31.6	
p2-m2 L	39.5	
p1-m2 L	42.8	
Depth p2	10.75	
Depth m1	12.5	
Notes.

L length

W width

Depth p2 and Depth 1m depth of mandible below p2 and m1, respectively

The p1 is represented by a single, shallow alveolus. The roots are preserved in the p2 alveoli. The p3 alveoli contain the distal tips of the roots. The p4 is the only tooth in the mandible with crown elements preserved (Fig. 1). The relatively intact roots of p2 and p4 are preserved, as are the apical-most portions of both of the roots of p3 and much of the anterior root of m1 (Figs. 1B and 1F). The main cusp of the p4 is broken anterior to the midline of the tooth. There is a distinct posterior accessory cusp on the external margin of the main cusp. The anterior cingulid is better developed antero-internally than antero-externally. The posterior cingulid is well developed.

The posterior root of the m1 is larger than the anterior root. The m2 is double rooted and situated on the slope of the ascending coronoid process. In the nano-CT volumetric reconstruction (Figs. 1B and 1F), the posterior root appears smaller than the anterior root. However, the bone surface surrounding that alveolus is eroded. The reconstruction in Fig. 2 is based on illustrations of Amphictictis schlosseri (Heizmann & Morlo, 1994) and A. milloquensis (Cirot & Wolsan, 1995), and casts of A. winterhofensis in the AMNH. As reconstructed, the two roots were nearly identical in depth, with the posterior alveolus extended upward and sloped toward the anterior margin of where the coronoid process would have been. Therefore, the m2 posterior root was a much more substantial structure that extended both distally and occlusally than what is preserved. There is a third small accessory rootlet between the anterior and posterior root.

Figure 2 Reconstruction of NCSM 33670 showing mandible with c1-m2.

Discussion

The presence of only a single, relatively non-diagnostic tooth, the p4, and the possibility that the jaw may not be late late Arikareean make a confident generic determination difficult. If an m3 were present, the root or its alveolus would have been evident in the microtomography scan (Fig. 1). The absence of an m3 indicates NCSM 33670 is a member of the Musteloidea—the Mephitidae, Ailuridae, Mustelidae, and Procyonidae (Wolsan, 1993; Wang, McKenna & Dashzeveg, 2005). The highly sloping distal molar, transitioning to the coronoid process out of the strict occlusal plane, is another feature seen only in musteloids, and immediately identifies the jaw as neither a canid nor an amphicyonid. An elongate m2 and an elongate m1 talonid (as suggested by the larger posterior m1 alveolus) are characteristic of the Ailuridae and Procyonidae. NCSM 33670 was therefore compared with late Oligocene and early Miocene basal musteloids, mustelids, ailurids, and procyonids, as well as with middle Miocene and younger ailurids and other musteloids.

NCSM 33670 compares best with Amphictis, the earliest and most primitive member of the Ailuridae (Morlo & Peigné, 2010). Amphictis is known from the late Oligocene (Arvernian, MP 28) to early Miocene (Orleanian, MN 3-4) of Europe (Heizmann & Morlo, 1994; Cirot & Wolsan, 1995; Morlo, 1996), the middle Miocene (late Orleanian, MN 5 or Astaracian, MN 6) of Turkey (Nagel, 2003), and the late early Miocene (early Hemingfordian, MN 4 equivalent) of North America (Tedford et al., 1987; Baskin, 2017). A possible earlier North American occurrence is AMNH 81029 (misnumbered AMNH 81040 in Baskin, 2017), a mandible from the late late Arikareean (AR4, MN 2-3 equivalent) of Nebraska identified as Bassariscus sp. (Cook & Macdonald, 1962). Baskin (2004) considered AMNH 81029 Amphictis-like.

The m1 length of NCSM 33670 is greater than that of Amphictis antiqua, A. borbonica. A. wintershofensis, and A. timicua; similar to that of A. ambigua, A. milloquensis, and A. schlosseri; and less than that of A. cuspida (Heizmann & Morlo, 1994; Cirot & Wolsan, 1995; Nagel, 2003; Baskin, 2017). The teeth have similar length proportions such as the ratios of p3, p4, and m2 to m1 (Table 2, Fig. 3). The middle Miocene (MN 6) A. cuspida from Çandir, Turkey is the latest and largest species (Nagel, 2003). It is known from a lower jaw with p4-m2. Amphictis cuspida differs from other Amphictis, including NCSM 33670, in having a shorter m2 as compared to the m1 (Table 2, Fig. 3). The only other specimen in Fig. 3 with such a relatively short m2 is the holotype of Plesictis milloquensis from the Late Oligocene (MP 29) of La Milloque in southwestern France (Helbing, 1928) which Cirot & Wolsan (1995) transferred to Amphictis. The additional specimen they identified as A. milloquensis (Cirot & Wolsan, 1995: Fig. 1) has the m2:m1 (Table 2) ratio the same as that of the early Miocene (MN 1–MN 2a) A. schlosseri from the Molasse basin and Mainz basin of southwestern Germany (Heizmann & Morlo, 1994). Amphictis from the Hemingfordian of Florida and Nebraska (Baskin, 2017) is much smaller than NCSM 33670. The Bassariscus-like posterior ramus with alveoli for the posterior root of m1 and for two roots of an elongate m2 from the Hemingfordian Pollack Farm of Delaware is similar to these Florida and Nebraska Amphictis (Emry & Eshelman, 1998; Baskin, 2003; Baskin, 2017). NCSM 33670 is near the high end of the alveolar measurements of the species of Amphictis plotted in Fig. 3. Measurements of Amphictis ambigua from Quercy at Pech du Fraysse (MP 28) are most similar to those of NCSM 33670. The p4 of NCSM 33670 is similar to that of A. schlosseri (Heizmann & Morlo, 1994: plate 2) in having well-developed anterior and posterior cingulids and an externally situated posterior accessory cusp. The posterior cingulid is wider in NCSM 33670.

Table 2 Measurements in mm of of taxa discussed in the text.

	n	p4 L	m1 L	m2 L	m2 L/m1 L	
NCSM 33670	1, 1, 1	8.1	10.8	6.9	0.68	
Amphictis antiqua1	?, 1, 1		9.3	5.9	0.63	
Amphictis.ambigua1	3, 5, 5	7.0 ± 0.45	10.2 ± 0.51	6.6 ± 0.56	0.65 ± 0.031	
Amphictis.milloquensis2	1, 1, 1	5.7	10.1	4.6	0.45	
Amphictis milloquensis3	1, 1, 1	5.0	10.1	5.8	0.57	
Amphictis borbonica5	1, 1, 1	6.2	9.0	5.2	0.58	
Amphictis borbonica1	?, 1, 1		9.4	5.2	0.55	
Amphictis schlosseri1	1, 3, 3	7.5	10.1 ± 0.15	5.9 ± 0.17	0.58	
Amphictis wintershofensis4	10,11, 10	6.3 ± 0.37	8.7 ± 0.47	5.8 ± 0.47	0.66 ± 0.035	
Amphictis prolongata5	1, 1, 1	6.5	8.8	6.0	0.68	
Amphictis cuspida6	1, 1, 1	8.5	12.6	5.8	0.46	
Amphictis timicua7	2, 2, 2	5.0, 5.3	7.5, 74	5.1, 50	0.68	
Actiocyon parverratis8	1, 1, 1	6.2	10.3	7.2	0.69	
Alopecocyon goeriachensis9	1, 1, 1	7.0	10.6	7.1	0.66	
Actiocyon sp.10	1, 1, 1	7.0	15.4	11.8	0.76	
Franconicits humilidens4	11, 16, 12	6.5 ± 0.32	8.9 ± 0.43	4.7 ± 0.31	0.54 ± 0.026	
Peignictis pseudamphictis11	?, 1, 1		5.3	2.5	0.47	
Floridictis kerneri7	10, 10, 3	7.5 ± 0.53	11.8 ± 0.37	2.8 ± 0.23	0.24	
Notes.

n number of specimens of p4, m1, m2

Sources of data:

1 Heizmann & Morlo (1994)

2 NMB LM 554, Cirot & Wolsan (1995)

3 FSP LM 1969 MC 2, Cirot & Wolsan (1995)

4 Dehm (1950)

5 Morlo (1996)

6 Nagel (2003)

7 Baskin (2017)

8 Smith, Czaplewski & Cifelli (2014)

9 Peigné (2012)

10 F:AM 25212, Ash Hollow Formation, Nebraska

11 De Bonis, Gardin & Blondel (2019)

Figure 3 Scatter plots comparing tooth dimensions of selected early and middle Miocene ailurids and procyonids.

Symbols are as follows: black star ⋆, NCSM 33670; black •, Amphictis ambigua1; black ►, Amphictis schlosseri1; black ▴, Amphictis milloquensis2; red •, Amphictis winterhofensis3; red ▴, Amphictis cuspida4; black ▾, Amphictis timicua5; red ▾, Amphictis cf. timicua5; red ■, Actiocyon parverratis5; black ■, Alopecocyon goeriachensis6; black ⧫, Bassaricyonoides phyllismillerae7; red ⧫, ?Edaphocyon palmeri7. Sources of data: 1Heizmann & Morlo (1994); 2Cirot & Wolsan (1995); 3Dehm (1950); 4Nagel (2003); 5Baskin (2017); 8Smith, Czaplewski & Cifelli (2014); 6Peigné (2012); 7Baskin (2003).

Alopecocyon from the middle Miocene of Europe (MN 6–MN 7/8) is closely related to Amphictis (De Beaumont, 1976; Morlo & Peigné, 2010). De Beaumont (1976) noted that the lower dentition of Alopecocyon goeriachensis, the type species of the genus, was difficult to distinguish from that of Amphictis. Wolsan (1993) synonymized Alopecocyon with Amphictis. Plots of dental measurements (Figs. 3 and 4) support Wolsan. However, Morlo & Peigné (2010) noted that this had not been accepted by subsequent authors. Webb (1969), following a suggestion from D. E. Savage, synonymized Actiocyon (Stock, 1947) from the Clarendonian Nettle Spring Fauna of the Caliente Formation of California with Alopecocyon. Smith, Czaplewski & Cifelli (2014) distinguished North American Actiocyon from Eurasian Alopecocyon by morphology of the cusps on m2. Actiocyon parverratis from the early Barstovian (MN 6 equivalent) of Nevada (Smith, Czaplewski & Cifelli, 2014) and Actiocyon sp. from the late Clarendonian Ash Hollow Formation of Nebraska have an elongate m2 relative to m1, but only the Ash Hollow specimen has the m2 significantly more elongate than Amphictis (Table 2). More derived ailurids such as Pristinailurus from the late Miocene to early Pliocene Gray Fossil Site in Tennessee (Wallace & Wang, 2004; Wallace, 2011) have a more complex p4 and an even more elongate m2. Tooth measurements (Table 2, Figs. 3 and 4) indicate NCSM 33670 is more similar to the larger species of Amphictis than to Alopecocyon or other ailurids.

Figure 4 Scatter plots comparing tooth dimensions of selected Amphictis and early and middle Miocene mustelids, and procyonids.

Symbols are as follows: black star ⋆, NCSM 33670; black •, Amphictis spp.1; red •, Amphictis winterhofensis2; black ▾, Amphictis timicua3; red ▾, Amphictis cf. timicua3; red ■, Actiocyon parverratis4; black ■, Alopecocyon goeriachensis5; black ⧫, Bassaricyonoides phyllismillereae6; red ⧫, ?Edaphocyon palmeri6; black ▴, Floridictis kerneri3; red ►, Zodiolestes3; black ◂, Oligobunis3; red ►, Brachypsalis3; black ◂, Promartes spp. 3; red ◂, Parabrachypsalis janisae3. Sources of data: 1Heizmann & Morlo (1994), Cirot & Wolsan (1995); 2Dehm (1950); 3Baskin (2017); 4Smith, Czaplewski & Cifelli (2014); 5Peigné (2012); 6Baskin (2003).

In North America, procyonids are known from the Hemingfordian of Florida, Nebraska, and Delaware. The procyonid from the early Hemingfordian Pollack Farm of Delaware (Emry & Eshelman, 1998) is an M1 compared to Edaphocyon (Wilson, 1960). The only mandible attributed to this genus, the Hemingfordian ?E. palmeri (Baskin, 2003), has an elongate m2 (Table 2), but differs from Amphictis and NCSSM 33670 in other dental proportions (Figs. 3 and 4). The Hemingfordian Bassaricyonoides phyllismillerae (Baskin, 2003) has p4 much wider especially posteriorly and with a more prominent posterior accessory cusp on the postero-external margin. The Barstovian to Recent Bassariscus and the Barstovian Probassariscus are smaller than NCSM 33670 and have a more elongate m2. In the eastern U.S. procyonids occur in Clarendonian and younger sites of Florida. The only extant procyonid that may have been present in North Carolina in the Pliocene and Pleistocene is the raccoon Procyon. In Procyon the m1 and m2 are more or less equal in length (Wright & Lundelius Jr, 1963) and the p4 is noticeably wider posteriorly and has a more laterally situated posterior accessory cusp.

Plesictis is a small mustelid from the late Oligocene and early Miocene of Europe (De Bonis, Gardin & Blondel, 2019). Mustelavus priscus (Clark, 1936) is a musteloid from the late Eocene of western North America (Baskin, 1998a). Simpson (1946, p.13) considered Mustelavus a “probable synonym” of Plesictis. Because of similarities with the type of M. priscus, Macdonald (1970), following Simpson, identified an anterior mandible from the early Arikareean (Ar2) of South Dakota as Plesictis sp. Tedford et al. (2004) noted this occurrence as an immigration event for Plesictis. However, the generic identity of this South Dakota specimen has not been demonstrated. Wang, McKenna & Dashzeveg (2005) include the Chadronian to Orellan Mustelavus priscus, Plesictis, and several other taxa as basal musteloids. The Orleanian Plesictis? humilidens (Dehm, 1950) is from Wintershoff-West. Wolsan (1993) made it the type species of Franconictis, which he classified as a mustelid that was the sister taxon of Stromeriella. Both have a relatively long m2. Franconicits humilidens has similar dimensions (Dehm, 1950: Table 11) to those plotted in Figs. 3 and 4 for Amphictis winterhofensis, other than a somewhat shorter m2 length relative to m1 (Table 2).

The early Oligocene Peignictis pseudamphictis is an enigmatic musteloid known from a posterior ramus with m1 (De Bonis, Gardin & Blondel, 2019). The posterior alveolus of the m2 is smaller than the anterior one. It is much smaller than Amphictis (Table 2) and has an m1 morphology more like that of Mustelictis (Bonis de, Gardin & Blondel, 2019). Although the m2 is relatively long, the m2:m1 ratio is less than that of Amphictis or NCSM 33670.

The Oligobuninae are the sister group of the neomustelids (Baskin, 1998a; Baskin, 2017; Valenciano et al., 2016). They, neomustelids, and procyonids are derived relative to ailurids in lacking an alisphenoid canal (Wang, McKenna & Dashzeveg, 2005). Promartes, the earliest oligobunine, is known from the late early Arikareean (Ar3) to Hemingfordian. Its species are similar in size to NCSM 33670 (Fig. 4). They and the other oligobunines such as Floridictis, Zodiolestes, Oligobunis, Brachypsalis, and Parabrachypsalis (Baskin, 2017) differ from NCSM 33670 in having a relatively shorter m2 and a relatively longer m1 (Table 2, Fig. 4).

Conclusions

The late Oligocene and early Miocene are marked by the immigration of carnivorans from Eurasia to North America (Tedford et al., 2004: fig. 6.3). Among the musteloids, Mustelictis from the early Oligocene (MP 22, 32 Ma) of Quercy, France (De Bonis, 1997) is one of the earliest stem mustelids (Wang, McKenna & Dashzeveg, 2005). Corumictis wolsani, from the early Arikareean (Ar1, MP 24 equivalent, 30–28 Ma and Ar2) of Oregon, is the earliest stem mustelid in North America and is an immigrant taxon (Paterson et al., 2020). Promartes is the only oligobunine mustelid known from the Arikareean. Whether the Oligobuninae are autochthonous or allochthonous is unresolved. Crown clade mustelids and procyonids first appear in North America during the Hemingfordian. The procyonids are derived from a European early Miocene taxon such as Broiliana (Baskin, 1998b). Previously the earliest ailurids in North America were from the Hemingfordian of Florida and Nebraska (Baskin, 2017). NCSM 33670 is most similar to the late Oligocene Amphictis ambigua (MP 28) or the early Miocene A. schlosseri (MN 1–MN 2a). Based on marine invertebrates, as noted above, the Belgrade Formation has been considered late Oligocene (Chattian or Chickasawhayan) to early Miocene (Aquitanian) in age. MacFadden et al. (2017) propose a late late Arikareean (AR 4, MN 2–3 equivalent) age assignment for the mammals from the Belgrade Formation, In any case, an Arikareean (late Oligocene to early early Miocene) age assignment supports NCSM 33670 being an immigrant from Eurasia and the earliest record of the Ailuridae in North America.

Supplemental Information

Supplemental Information 1 Ailuridae dental measurements used to construct Figure 3

Click here for additional data file.

Supplemental Information 2 Mustelida dental measurements used to construct Figure 4

Click here for additional data file.

We thank Donald Clements for information on the stratigraphy of the region and providing assistance to John Dubois. Bruce MacFadden supplied information on the ages of the terrestrial vertebrates from the mine. We are grateful to Shruti Kolli and Abigail Malach for assistance with scanning of the specimen. Access to the AMNH collections was granted by Drs. Richard Tedford and Jin Meng, and assistance was provided by Judith Galkin. We thank M. Morlo, M. Wolsan, and an anonymous reviewer for the constructive comments that improved the manuscript.

Institutional abbreviations

AMNH American Museum of Natural History, New York, New York

F:AM Frick Collection of fossil mammals in the AMNH

NCSM North Carolina Museum of Natural Sciences

Anatomical Abbreviations

p lower premolar

m lower molar

L length

W width

Additional Information and Declarations

Competing Interests

Author Contributions

Field Study Permissions

Data Availability

Henry Galiano is the owner of Maxilla & Mandible, Ltd., New York, New York.

Jon Baskin conceived and designed the experiments, performed the experiments, analyzed the data, prepared figures and/or tables, authored or reviewed drafts of the paper, and approved the final draft.

Edwin Dickinson and Adam Hartstone-Rose performed the experiments, analyzed the data, prepared figures and/or tables, authored or reviewed drafts of the paper, and approved the final draft.

John DuBois conceived and designed the experiments, performed the experiments, authored or reviewed drafts of the paper, and approved the final draft.

Henry Galiano performed the experiments, analyzed the data, authored or reviewed drafts of the paper, drew Figure 2 and approved the final draft.

The following information was supplied relating to field study approvals (i.e., approving body and any reference numbers):

Permission to collect was granted by the operators of the Martin Marietta quarry at Belgrade, North Carolina.

The following information was supplied regarding data availability:

The raw measurements used to construct Figures 3 and 4 are available in the Supplemental Files.

NCSM 33670 is accessioned in the North Carolina Museum of Natural Sciences, Raleigh, North Carolina. AMNH specimens are available in the collection of the Department of Paleontology, American Museum of Natural History, New York, New York (accession numbers: AMNH 81029, AMNH 81040).

The 3D model is available at Morphosource: https://www.morphosource.org/Detail/ProjectDetail/Show/project_id/973.

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
