# Peer review of "?Amphictis (Carnivora, Ailuridae) from the Belgrade Formation of North Carolina, USA"

_PeerJ, doi:10.7717/peerj.9284_

## Round 0.1 · original submission · Minor Revisions

While this is a very incomplete specimen that would not be considered publishable by some (one of the reviewers), I do agree with the reviewers that its unusual nature makes it publishable, together with your very careful and tentative approach to identifying and discussing it. The paper is short and to the point.

Critical comments that need to be addressed by reviewer 1 are:
You should explain on what the reconstructed teeth in Fig 2 are based on; clarify the statement regarding the absence of m3; explain why not all species in the genus are discussed. Elucidate the statement regarding the similarity of p4 to A. schlosseri.

Critical comments by Reviewer 2:
Consider expanding comparisons to other genera as listed; change the species indication to reflect female genus name; provide quantified ratios for tooth shapes.

·

Basic reporting

The manuscript is well structured and written in understandable English even for non-native English speakers.
Introduction, background information, and figures are fine, but
- I suggest including a map showing all the localities mentioned in the introduction. Moreover, geographical coordinates for the Belgrade Quarry should be provided.
- Figure 2 shows a reconstruction of the specimen, but it is unclear which source the authors used for the reconstructed teeth (Amphictis ambigua? A. schlosseri?).
- The new specimen is referenced in figure 3 and 4 with two different colors (black/red). This should be unified.
Literature is well referenced and relevant, but
- I noticed that the journal’s format is not always followed. I commented this in the pdf, but please check again for consistency.
I additionally
- corrected some very minor misspellings within the pdf.

Experimental design

The manuscript fits well to aims and scope of the journal. The applied methods are described with sufficient detail.
All other criteria of this section are fulfilled, as well.

Validity of the findings

The results are of high interest as the authors verify the discussed specimen as the yet oldest ailurid of North America. Due to this, I expect a high rate of citations not only in literature of specialist but also in general contributions on Ailuridae, up to Wikipedia.
I have only few additional remarks:
- the authors state that Morlo & Peigné (2010) discuss “all eight species” of the genus. In figures 3-4, however, only five of these are included, but it remains unclear, why that is the case.
Moreover, the species A. timicua Baskin, 2017 should be added to the eight species mentioned by Morlo & Peigné (2010).
- the authors state p4 to be closest to A. schlosseri, but it is unclear why they say this. I suggest to include the relevant characters.
- the authors state m3 to be absent. This is very probably true, but in my view this cannot be evidenced from the specimen or the figures attached to this manuscript. The authors should either add such evidence (lack of the anterior border of a possible m3-alveolus?) or clarify that the absence of m3 is inferred from the other evidence that allows identification of the specimen as musteloid.

Additional comments

This contribution deals with a new mandibular ramus from the early Miocene of the Belgrade Quarry. Though only little information is obtainable from the specimen, the authors attribute it convincingly to the genus Amphictis, verifying it to be the hitherto oldest member of the carnivoran family Ailuridae in North America.
This manuscript thus is an important contribution to the evolution and early paleobiogeography of ailurids. I suggest to publish it with the minor changes I outlined above.
I thank you for the possibility to be part of its publication as a reviewer.

Reviewer 2 ·

Basic reporting

- I am not a native English speaker thus I cannot comment on the language
- The geological context is well brought forward. The literature is referenced but some citations are missing
- extend comparisons

Experimental design

The study shows that a scan method may be used for an uncomplete fossil material to help a taxonomic identification. It is in the scope of the journal. The research question is well defined.
The research is rigorous with a high technical standard and the method well described.

Validity of the findings

Despite the poor material, I think it is valid and could contribute to the knowledge on geographic distribution of a taxon quite rare in North America.
The conclusion is credible but the comparisons could be a little extended

Additional comments

This is an original and smart analysis on an isolated and almost edentulous hemimandible of carnivoran from North Carolina of which only a piece of p4 is known. I appreciate the idea to work with help of microtomography scans to use the shape and size of the roots for the identification.
The authors give a careful description of the fossil and they use a large comparison to conclude that the hemimandible belongs probably to ?Amphictis. Measurements in the supplemental will be very useful.
Nevertheless, I have some remarks
- Currently, the suffix ictis coming from a feminine Latin word, the species name Amphictis ambigua must be used.
- Ratios are used in the text but only as larger or smaller, please give the numbers.
- line 193: the authors speak about the measurements of Pech du Fraysse. Those of m2 were published (Cirot & Morlo 1996) but where did they take those of m1? (Citation please).
- For the comparisons, some other taxa may be added. First, I think to Pseudobassaris. it is considered as a procyonid by Wolsan & Lange-Badré 1996, who attributed to it a couple of teeth, m1 and m2, from two Quercy localities (Bonis & Cirot 1995 fig. 8-9). If they are right, although they come from two different localities, the ratio Lm2/Lm1 (65) is close to that of NCSM 33670 (63). The size is smaller but the hypothesis is possible.
- Two other genera, Broiliana and Stromeriella could also be candidate, although they are smaller than NCSM 33670.
- Finally, I don’t understand why, in figure 4, there are genera which are not discussed in the text: Floridictis, Zodiolestes, Oligobunis, Brachypsalis, Promartes, Parabrachypsalis, even if they are in the measurements.
- line 527: Cirot & Wolsan 1996 no Cirot & Bonis

The fossil material is poor, but it is another point in North America where the presence of this type of carnivoran is recorded (even with a double question mark)

·

Basic reporting

no comment

Experimental design

no comment

Validity of the findings

no comment

Additional comments

The fossil described in this manuscript is very incomplete, and its geologic age is not certain, which can only result in an approximate taxonomic assignment and likely potential trouble for subsequent authors. One could therefore wonder if this manuscript should be published at all. As concern myself, I avoid publication of specimens that are such incomplete anatomically and uncertain taxonomically. However, I admit that there may be some importance in this fossil for an American or other reader that I missed. Therefore, I recommend publication of this manuscript for the sake of record.

That said, despite a very incomplete state of the evaluated fossil, I feel convinced by the authors that the fossil may be tentatively referred to Amphictis. As concerns the manuscript itself, I find it professional and well written. I believe this manuscript is publishable in the present form. I have only one minor comment that Amphictis ambiguus should be spelled as Amphictis ambigua because ictis (weasel in Latin) is feminine in gender.

Typos:
Line 84: “Arikareean, to”
Line 134: “missing, Length”
Line 162: “m2 is and”
Line 187: “basinsof”
Line 215: “the the”
Line 216: “PollackFarm”
Line 274: “anycase”
Line 527: “Cirot and Bonis, 1995”

---

## Round 0.2 · accepted · Accept

Thank you for taking the concerns of the reviewers seriously, this manuscript looks good. I am pleased that pandas are making it into PeerJ!